# Balancing Innovation and Safety: Prediction, Prevention, and Management of Pneumonitis in Lung Cancer Patients Receiving Novel Anti-Cancer Agents

**DOI:** 10.3390/cancers17152522

**Published:** 2025-07-30

**Authors:** Sarah Liu, Daniel Wang, Andrew Robinson, Mihaela Mates, Yuchen Li, Negar Chooback, Pierre-Olivier Gaudreau, Geneviève C. Digby, Andrea S. Fung, Sofia Genta

**Affiliations:** 1Department of Medicine, Division of Respirology and Sleep Medicine, Queen’s University, Kingston, ON K7L 3N6, Canada; 15ssl5@queensu.ca (S.L.);; 2Faculty of Medicine, University of Ottawa, Ottawa, ON K1H 8M5, Canada; 3Department of Oncology, Queen’s School of Medicine, Queen’s University, Kingston, ON K7L 3N6, Canada; 4Cancer Centre of Southeastern Ontario, Kingston Health Sciences Centre, Kingston, ON K7L 2V7, Canada; 5Arthur J.E. Child Comprehensive Cancer Centre, Department of Oncology, University of Calgary, Calgary, AB T2N 4N1, Canada; andrea.fung@albertahealthservices.ca

**Keywords:** lung cancer, pneumonitis, antibody–drug conjugate, bispecific antibody, immunotherapy toxicity

## Abstract

Pneumonitis is a potential toxicity of common anti-cancer therapies, including radiotherapy, chemotherapy, immunotherapy, and targeted therapy. Due to underlying disease and comorbidities, lung cancer patients are at high risk of developing fatal cases of pneumonitis or permanent impairment of their respiratory function. Novel anti-cancer agents, including antibody–drug conjugates and bispecific antibodies, have recently been incorporated into clinical use for lung cancer patients due to their demonstrated antitumor activity. However, these medications may be associated with an increased probability of pneumonitis; therefore, implementation of strategies to optimize risk stratification and management of pneumonitis is urgently needed to improve patient care.

## 1. Introduction

Lung cancer is the leading cause of cancer-related deaths worldwide, with around 2.48 million new cases and 1.8 million deaths annually [1]. In Canada, it is estimated that 32,100 individuals will be diagnosed with lung cancer in 2024, and approximately 20,700 Canadians will die from the disease [2]. Lung cancer is primarily classified into non-small cell lung cancer (NSCLC) and small cell lung cancer (SCLC). NSCLC (e.g., adenocarcinoma, squamous cell carcinoma, and large cell carcinoma) accounts for around 85% of lung cancer cases. On the other hand, SCLC is a more aggressive form of lung cancer, which tends to spread rapidly to distant organs, and accounts for approximately 15% of lung cancer cases [3]. Major risk factors for both NSCLC and SCLC include cigarette smoking, second-hand smoke, occupational exposure, air pollution, and genetic susceptibility. Currently, lung cancer is diagnosed through imaging and pathological assessment of tissue biopsies [3]. Low-dose CT scans are typically used for early detection in high-risk individuals, while biopsy remains the gold standard for a definitive diagnosis [4].

Pneumonitis, a localized or widespread inflammation of the lung parenchyma, is a well-documented adverse effect of several anti-cancer therapies, including radiotherapy, chemotherapy, immune checkpoint inhibitors (ICI), and targeted therapies [5,6,7]. The incidence and severity of pneumonitis vary depending on cancer type and patient characteristics, such as age, sex, and comorbidities [8,9]. The identification and treatment of pneumonitis is particularly challenging in patients with lung cancer, as its clinical manifestations often mimic those of disease progression or pulmonary infection, making it difficult to determine the best management [10]. This resemblance, coupled with a greater prevalence of pre-existing respiratory impairment, can hinder the diagnosis of pneumonitis and subsequent intervention. Moreover, due to a higher rate of compromised respiratory function, lung cancer patients face a heightened risk of developing more severe and fatal cases of pneumonitis [11]. In fact, pneumonitis is the leading cause of therapy-related mortality in lung cancer patients, with an early mortality rate of 36% [11].

The type of anti-cancer treatment received has an impact on the risk of developing pneumonitis. The incidence of pneumonitis with conventional therapies currently utilized in lung cancer treatment is summarized in Table 1. For instance, pneumonitis occurs in 15–40% of non-small cell lung cancer patients within one to six months post-radiotherapy, whereas this toxicity only occurs in approximately 9.5% of NSCLC patients treated with immune checkpoint inhibitors [5,12,13,14]. The rate of pneumonitis among patients treated with targeted therapies is generally lower. For example, the overall incidence of pneumonitis among NSCLC patients treated with epidermal growth factor receptor (EGFR) tyrosine kinase inhibitors is around 1% [15]. Novel anti-cancer therapies such as antibody–drug conjugates (ADCs) and bispecific antibodies (BsAbs) are rapidly becoming a standard of care in multiple oncology settings, including lung cancer; however, these agents are also associated with potentially life-threatening toxicities, including pneumonitis [16,17,18].

The pathophysiological factors underlying pneumonitis risk may vary across different classes of anti-cancer agents, underscoring the need for further mechanistic studies. A deeper understanding of the mechanism of action of novel agents and the impact of patient characteristics (e.g., comorbidities such as underlying pulmonary conditions) is crucial for predicting toxicities and refining treatment strategies to improve the overall safety and efficacy of lung cancer therapies. Efforts should be made to define risk factors associated with pneumonitis so that patients at risk of developing pneumonitis can be identified early and monitored closely during treatment.

This narrative review aims to (1) examine the incidence of pneumonitis associated with conventional and novel anti-cancer therapies, particularly ADCs and BsAbs, (2) elucidate potential pathophysiological mechanisms underlying pneumonitis risk, and (3) evaluate emerging predictive biomarkers, which could help minimize future pneumonitis risk for lung cancer patients. A health sciences librarian was consulted, and a literature search was completed utilizing Medline, Embase, PubMed, and Google Scholar databases.

## 2. Novel Anti-Cancer Agents

The incidence of treatment-related pneumonitis associated with novel agents such as ADCs and BsAbs is summarized in Table 2. The following sections highlight the pneumonitis risk reported with specific drugs within each class of novel therapies. 

### 2.1. Antibody–Drug Conjugates

Antibody–drug conjugates are a class of anti-cancer agents that consist of a monoclonal antibody covalently attached to a cytotoxic drug via a chemical linker (Figure 1) [34]. This combines a highly specific targeting ability with a potent killing effect to achieve accurate elimination of cancer cells [34]. Upon binding of the monoclonal antibody to the target antigen, the ADC is first internalized into the tumor cell. The linker molecule then breaks down to promote intracellular release of the payload, where it exerts its cytotoxic effects [35].

ADCs have had marked success in the treatment of solid and hematological malignancies, including lymphomas, breast, lung, gastrointestinal, and urothelial cancers since the first approval of gemtuzumab ozogamicin by the Food and Drug Administration in 2000 [36]. However, this class of agents is also associated with potential side effects. Importantly, all three key components of ADCs—the antibody, the cytotoxic payload, and the linker—have a critical impact on the toxicity profile of these therapies [37].

Pneumonitis is a potentially life-threatening adverse event that has been associated with multiple ADCs targeting various antigens [36,37]. The factors contributing to ADC-induced pneumonitis are not completely understood, but hypothesized mechanisms include (1) expression of the ADC target antigen in normal lung tissue; (2) instability of the linker leading to premature release of cytotoxic drugs into the bloodstream; (3) high drug-antibody ratios; (4) bystander effect of ADCs (in which cells nearby tumor cells are killed by the cytotoxic payload regardless of whether they are antigen positive or negative); and (5) drug–drug interactions (Figure 2) [17,37].

Drug-related pneumonitis of all grades has been shown to occur with the large majority of ADC therapies used for cancer treatment, including trastuzumab emtansine (3.62%), enfortumab vedotin (5.24%), brentuximab vedotin (3.22%), polatuzumab vedotin (2.56%), gemtuzumab ozogamicin (2.53%), and inotuzumab ozogamicin (2.33%), although to a lesser extent than trastuzumab deruxtecan (T-DXd) [38]. However, it must be noted that several of these ADCs were studied in patients with hematologic cancers such as lymphoma and leukemia as opposed to solid tumor cancers, which may result in difficulty comparing rates of pneumonitis. In the next sections, we will report available data for ADCs specifically investigated in lung cancer patients.

#### 2.1.1. Anti-HER2 ADCs

Human epidermal growth factor receptor 2 (HER2) is a known oncogenic driver and potential therapeutic target in several types of tumors, including NSCLC [39]. HER2 mutations, overexpression, and amplifications have been reported in 1–7%, 8–23%, and 2–5% of NSCLC cases, respectively [40].

**Trastuzumab deruxtecan (T-DXd)** is a HER2-directed ADC used in the treatment of HER2-positive advanced solid tumors, including breast cancer, gastric cancer, and NSCLC [16]. T-DXd has a high drug-to-antibody ratio (DAR) of 8:1. Pneumonitis is recognized as a significant adverse event related to T-DXd. The incidence of all-grade pneumonitis in patients treated with T-DXd ranges from 10.5% to 20% in various studies, with grade 3 or higher events occurring in up to 6% of patients [16,21,41,42,43]. Fatal cases have been reported, emphasizing the need for vigilant monitoring [44].

The probability of developing T-DXd-induced pneumonitis may be influenced by several factors, including the dose of the ADC and the site of the primary tumor. For instance, in a phase 2 study including 184 patients treated with T-DXd (5.4 mg/kg) for HER2-positive metastatic breast cancer, the incidence of drug-related pneumonitis was reported to be 13.6% for all grades, with grade 3 or higher events occurring in approximately 2.7% of patients, including four deaths felt to be attributable to pneumonitis [17]. The DESTINY-Lung01 phase 2 multicentre trial enrolled a total of 49 HER2-overexpressing NSCLC patients in cohort 1 (6.4 mg/kg of T-DXd) and reported an incidence of 20% for all-grade pneumonitis, with grade 3 or higher events in 6% of patients, including three deaths attributable to drug-related lung disease [41]. Of the 41 HER2-overexpressing NSCLC patients enrolled in cohort 1A (evaluating 5.4 mg/kg of T-DXd), only 7.3% were found to have pneumonitis, with 4.9% grade 3 or higher events (including two deaths), suggesting a possible dose-related effect contributing to the development of adverse events like pneumonitis. Furthermore, in cohort 2 of DESTINY-Lung01, T-DXd (6.4 mg/kg) was used for the treatment of *HER2*-mutant NSCLC patients, and drug-related pneumonitis occurred in 26% of the 91 patients enrolled, with grade 3 or higher events in 6.6% of patients [21]. The phase 2 DESTINY-Lung02 study compared the 5.4 mg/kg vs. 6.4 mg/kg dosing of T-DXd in *HER2*-mutant NSCLC patients and confirmed a higher incidence of drug-related pneumonitis with the higher dose: 12.9% vs. 28.0% of patients had reported drug-related pneumonitis (2.0% grade ≥ 3 in each arm) in the 5.4 mg/kg and 6.4 mg/kg groups, respectively [22].

**Trastuzumab emtansine (T-DM1)**, an ADC with a microtubule inhibitor payload, is also a HER2-directed ADC primarily used in the treatment of HER2-positive breast cancer. T-DM1 has a DAR of 3.5:1. In NSCLC, T-DM1 has shown some efficacy in HER2-overexpressing and *HER2*-mutant cases [45,46]. The incidence of drug-induced pneumonitis observed with T-DM1 is relatively low. In a phase 3 trial investigating T-DM1 (3.6 mg/kg every 3 weeks) in HER-2-positive breast cancer patients, pneumonitis of any grade occurred in 2.6% of the 743 patients treated with T-DM1 [42]. In a smaller trial investigating T-DM1 in patients with NSCLC, 49 patients were treated with 3.6 mg/kg every 3 weeks, and 0% developed pneumonitis [43].

#### 2.1.2. Anti-HER3 ADCs

Human epidermal growth factor receptor 3 (HER3) is a protein known to be hyper-expressed in *EGFR*-mutated lung cancer and a mediator of resistance to tyrosine kinase inhibitors (TKIs) [47,48].

**Patritumab-deruxtecan (HER3-DXd)** is an HER3-targeting ADC characterized by a topoisomerase I inhibitor payload and a cleavable linker [49]. Similar to T-DXd, HER3-DXd has a DAR of 8:1. In the HERTHENA-Lung01 phase 2 study, a total of 225 patients with *EGFR*-mutated NSCLC previously treated with TKIs received HER3-DXd at a dose of 5.6 mg/kg once every 3 weeks [23]. In this EGFR inhibitor-resistant population, HER3-DXd resulted in an objective response rate (ORR) of 29.8%, and a median progression-free survival (PFS) and overall survival (OS) of 5.5 and 11.9 months, respectively. Pneumonitis of any grade was identified in 12 patients (5.3%), including two cases of grade 3 pneumonitis and one case of fatal lung involvement. Similarly, the HERTHENA-Lung02 phase 3 study evaluating HER3-DXd vs. platinum-based chemotherapy in *EGFR*-mutant NSCLC after disease progression on a third-generation TKI reported adjudicated drug-related ILD in 5% of patients (n = 14; 11 grade 1/2, 1 grade 3, and 2 grade 5) in the HER3-DXd arm [24].

#### 2.1.3. Anti-TROP2 ADCs

Trophoblast cell surface antigen 2 (TROP2) is a transmembrane protein overexpressed in several cancer types, including 42–60% of NSCLC cases [50], with minimal expression in normal lung tissue [51].

**Datopotamab deruxtecan (Dato-DXd)** is an ADC-targeting TROP2, consisting of a humanized anti-TROP2 monoclonal antibody linked to a topoisomerase I inhibitor payload via a cleavable linker [25]. In the phase 3, randomized TROPION-Lung01 study, 604 patients with pretreated advanced/metastatic NSCLC were randomized to receive Dato-DXd or docetaxel. This study found that Dato-DXd significantly improved progression-free survival as compared to docetaxel (4.4 months compared to 3.7 months) [25], with a greater benefit from the ADC observed in the non-squamous histology subgroup (median PFS 5.5 months in the Dato-DXd group vs. 3.6 months in the docetaxel group). Importantly, the incidence of any-grade pneumonitis was 8.8% in patients treated with the ADC versus 4.1% in patients treated with docetaxel, with grade 3 or higher pneumonitis occurring in 2.7% of patients vs. 1.4%. Similarly, in the phase 1 single-arm TROPION-PanTumor01 study, the incidence of pneumonitis was 6% among 50 patients receiving Dato-DXd at the recommended dose of 6 mg/kg, with grade 3 or higher pneumonitis reported in 2% of patients (no grade 5 events) [26]. Interestingly, there appeared to be a higher incidence of adjudicated drug-related pneumonitis in the 8 mg/kg dose arm (13.8% any grade, 5% grade 3 or higher), with three grade 5 events reported with the higher dose.

As previously reported for HER2 targeting ADCs, there appears to be a difference in the rates of drug-induced pneumonitis in patients treated with Dato-DXd who have lung cancer compared to those who have other tumor types, such as breast cancer. In a pooled analysis of 927 NSCLC patients from the TROPION-Lung01, -Lung05, and -PanTumor01 studies who received Dato-DXd, the incidence of any-grade pneumonitis was 8.8%, with grade 3 or higher pneumonitis occurring in 2.7% of patients [52]. In contrast, results from the TROPION-PanTumor01 phase 1 trial demonstrated that in patients with advanced/metastatic hormone-positive/HER2-negative and triple-negative breast cancer, the incidence of any-grade pneumonitis was 5%, with grade 3 or higher pneumonitis occurring in 1.7% of patients [53].

**Sacituzumab govitecan (SG)** is another TROP2-targeting ADC, comprising a topoisomerase I inhibitor payload and a cleavable linker. Similar to Dato-DXd, SG has been compared to docetaxel in the phase 3 randomized EVOKE-01 clinical trial evaluating 603 previously treated NSCLC patients [54]. Although no significant difference in overall survival was observed in the intention-to-treat (ITT) population, an improvement in overall survival of 3.5 months was observed in the subgroup of patients who did not respond to prior immunotherapy. The authors reported one case of fatal pneumonitis in the docetaxel arm, while no cases of pneumonitis were reported among the patients treated with the ADC. SG is also being tested in combination with pembrolizumab as first-line treatment for PD-L1 high NSCLC patients in the phase 2 EVOKE-02 study [55]. Preliminary results for the first 30 patients showed an objective response rate of 67% [55]. While no specific data for pneumonitis were reported, grade 3 or higher treatment-emergent adverse events, including respiratory failure, were observed in 10% of the patients. SG has also been tested for the treatment of other cancer types, including breast and endometrial cancer. Notably, no cases of SG-related pneumonitis were reported in phase 2 and 3 clinical studies in these settings [27,56]. Structural differences between SG and Dato-DXd may explain the different toxicity profiles observed for these two ADCs. While both compounds include a topoisomerase I inhibitor payload, deruxtecan is more potent and has a longer half-life compared to govitecan. Furthermore, while SG uses a hydrolyzable linker, Dato-DXd employs a protease-cleavable linker designed to be activated in the lysosome [57].

### 2.2. Bispecific Antibodies

Bispecific antibodies are another class of novel anti-cancer agents currently being explored for the treatment of a variety of advanced cancers. BsAbs are engineered molecules designed to bind two different antigens or epitopes simultaneously. They consist of two distinct antigen-binding sites, which can engage both tumor-associated antigens and/or immune cell receptors (Figure 3). One type of BsAb is a bispecific T-cell engager, which targets a tumor antigen and CD3 on T cells, thereby facilitating the immune system’s ability to target and destroy cancer cells through T-cell activation and subsequent tumor cell lysis [58,59].

BsAbs have shown efficacy in various cancers, including hematologic malignancies and solid tumors. For instance, blinatumomab, a BsAb targeting CD19 and CD3, is approved for B-cell precursor acute lymphoblastic leukemia [60]. In solid tumors, BsAbs are being explored for their potential to target specific tumor antigens, such as HER2 in breast cancer and delta-like ligand 3 (DLL3) in SCLC [58,59]. **Tarlatamab**, a bispecific T-cell engager that targets CD3 and DLL3 is the first T-cell engager approved for the treatment of SCLC. In the DeLLphi-300 open-label phase 1 study of 107 patients with relapsed/refractory SCLC, tarlatamab had an objective response rate (ORR) of 23.4%, a median PFS of 3.7 months, and a median duration of response of 12.3 months [28]. In this study, the incidence of any-grade pneumonitis was 4.7% (n = 5), with one patient suffering a grade 5 pneumonitis [28]. In the phase 2 DeLLphi-301 trial, tarlatamab was evaluated in 220 patients with previously treated SCLC. The incidence of treatment-related pneumonitis in the overall population was approximately 1.4% [29,61]. In the recently published phase 3 DeLLphi-304 trial, tarlatamab (10 mg every 2 weeks) was associated with an overall survival benefit when compared to chemotherapy (13.6 months vs. 8.3 months) in the second-line treatment of extensive-stage SCLC patients. There were no reported cases of pneumonitis observed in the 254 patients treated with tarlatamab; the most common adverse event was low-grade cytokine release syndrome [62].

Other BsAbs targeting different tumor antigens, such as HER2 and EGFR, are under investigation for their potential use in lung cancer [63]. One example is **zanidatamab**, a BsAb targeting two non-overlapping HER2 domains. In early-phase trials, zanidatamab has shown promising efficacy in HER2-expressing NSCLC, with an ORR of approximately 33% in heavily pretreated patients. In a phase 1 study by Meric-Bernstam et al., zanidatamab demonstrated a manageable safety profile among 46 patients, with common adverse events being diarrhea and infusion reactions, but there were no significant reports of pneumonitis [30]. Another example is **amivantamab**, which targets both EGFR and cMet. This BsAb has demonstrated efficacy in *EGFR*-mutant NSCLC, particularly in patients with resistance to EGFR TKIs [63,64]. In clinical trials, amivantamab achieved an ORR of 36% in patients with EGFR exon 20 insertion mutations. The Phase I CHRYSALIS trial, which evaluated amivantamab in 362 patients with EGFR exon 20 insertion-mutated NSCLC, reported an incidence of 3.3% for all-grade pneumonitis, with 0.7% of patients experiencing grade 3 or higher events [31,65]. Amivantamab has also shown promising results in metastatic NSCLC patients with common activating EGFR mutations, including use in the first-line setting in combination with lazertinib (a 3rd generation EGFR TKI) as studied in the MARIPOSA trial [66], as well as use in later lines in combination with chemotherapy +/− lazertinib (MARIPOSA-2) following disease progression on osimertinib [67]. Interstitial lung disease or pneumonitis rates were typically low, with an incidence of 3% in patients treated with amivantamab plus lazertinib (1% grade 3 or higher) in the MARIPOSA study [66]. Similarly, approximately 1% of patients developed pneumonitis with the combination of amivantamab plus platinum doublet chemotherapy and 3% with amivantamab plus chemotherapy plus lazertinib in the MARIPOSA-2 trial [67]. Another promising approach in the treatment of *EGFR*-mutated NSCLC is the dual targeting of EGFR and HER3. **Izalontamab** (SI-B001) and **BL-B01D1** are two examples of anti-HER3/anti-EGFR BsAbs currently under investigation in early-phase clinical trials [32,33]. While no drug-induced lung adverse events have been reported for Izalontamab so far, one case of pneumonitis was reported in a phase 1 clinical trial testing BL-B01D1 in 195 patients with advanced solid tumors, including 113 NSCLC patients [32].

Another therapeutic strategy currently being explored via BsAbs is the simultaneous inhibition of two distinct immune checkpoints. **Rilvegostomig** (anti-PD1/anti-TIGIT BsAb), **Volrustomig** (anti-PD1/anti-CTLA4 BsAb), and **Tebotelimab** (anti-PD1/anti-LAG3 BsAb) are some examples of dual immune checkpoint inhibitors currently under investigation in patients with advanced solid tumors, including lung cancer [68,69,70]. While toxicity data for these agents are not fully published yet, the known risk of pneumonitis associated with monoclonal antibodies targeting PD-(L)1 foreshadows the possibility of increased toxicity rates for patients treated with dual inhibitors.

## 3. Risk Factors and Predictive Biomarkers of Treatment-Related Pneumonitis

Most commonly, treatment-induced pneumonitis is relatively minor and will resolve with appropriate therapy or after discontinuation of the offending agent. Nevertheless, in a small subset of patients, this toxicity can have severe consequences, including permanent impairment of respiratory function, significant symptom burden, requirement for supplemental oxygen therapy, and even death. Over the past decade, the increase in the number of patients receiving immune checkpoint inhibitors and targeted therapies has led to a parallel increase in the incidence of treatment-related adverse events [71]. As a result, considerable effort has been made to develop biomarkers predicting the development of treatment-induced toxicities, including pneumonitis [5,72].

Identification of patients with a higher risk of developing pneumonitis in advance of treatment start can be challenging, as multiple factors can contribute to the etiopathogenesis of this complication. Several studies indicate that underlying patient conditions, such as smoking status or pre-existing lung disease, including COPD, can predispose patients to the development of pneumonitis [73]. Similarly, tumor-specific factors, such as squamous histology and high PD-L1 expression, have been correlated with a higher risk of pneumonitis in NSCLC patients [74].

Novel biomarkers are being evaluated to determine their utility in predicting treatment-related pneumonitis in cancer patients (Table 3). Multiple studies indicate that single-nucleotide polymorphisms of genes regulating processes such as inflammation and DNA repair may represent a useful tool to predict radiation-induced pneumonitis in lung cancer patients [75,76,77,78,79]. Aside from their direct utility in identifying groups of patients at higher risk of toxicity, these observations may also serve as a warning regarding the potential risks of combination radiotherapy with drugs targeting DNA repair inhibitors or immunotherapies.

While meaningful data are lacking for patients receiving chemotherapy and tyrosine kinase inhibitors, several blood biomarkers have been tested or are under development for the early identification of cases of treatment-related pneumonitis in patients receiving immunotherapy. These include immune-cytokines [80,81], autoantibody panels [85,86], and the presence of specific HLAs subtypes, in particular certain HLA-B alleles such as HLA-B*35:01 [83,84]. Importantly, commonly measured parameters such as C-reactive protein and white blood cell counts have also demonstrated a possible correlation with the risk of immune-related toxicities; for example, a low neutrophil count and significant baseline elevation of eosinophil count have both been associated with a higher probability of immune checkpoint inhibitor pneumonitis [88]. T cell receptor (TCR) metrics are another factor currently being studied as potential predictors of both anti-cancer efficacy and risk of toxicity in cancer patients receiving ICI [87]. In a study published by Murray et al., paired analysis of TCR sequencing done at baseline and on treatment from peripheral blood collected from patients treated with an anti-PD1 agent demonstrated a dynamic reshaping of the TCR repertoire in patients experiencing immune-related adverse events (irAEs) [87]. In particular, patients who developed immune-related toxicities (including pneumonitis) displayed a greater proportion of both expanding and regressing TCR clones while on treatment. Importantly, these changes were detectable 6 weeks after treatment initiation, underlining the potential utility of TCR parameters as an early predictor of irAEs.

Although data for novel anti-cancer therapies such as ADCs and BsAbs are still limited, their known immunomodulatory effects suggest that blood and tissue-based biomarkers predictive of pneumonitis from ICI may also be applicable to these agents [93]. Certainly, the expression of the targeted antigen in lung tissue plays a significant role in determining the risk of pulmonary toxicity, and is therefore a factor that should be taken into consideration when designing a novel anti-cancer compound.

Another promising non-invasive approach to predicting the risk of pneumonitis in cancer patients is represented by radiomics [89,90]. This type of analysis allows the extraction and characterization of quantitative data such as shape, texture, and intensity features from standard radiological imaging to derive biological information on the composition of body tissues, including cancer [91]. Encouraging data supporting the utility of radiomic parameters as predictive biomarkers have been described for both ICI and radiation-induced pneumonitis [89,90,92]. Importantly, radiomic techniques are rapidly becoming a key resource in precision oncology, due to their ability to investigate not only treatment-related toxicities, but also tumor aggressiveness and sensitivity to potential therapeutic options. While the role of radiomics in predicting the risk of pneumonitis from novel anti-cancer agents has not been established yet, the utility of this approach in predicting responses to ADCs is starting to be explored [94].

## 4. Management of Treatment-Related Pneumonitis and Treatment Rechallenge Following Drug-Induced Pneumonitis

The initial management of treatment-related pneumonitis is similar across different anti-cancer therapies and typically involves dose interruption, reduction, or treatment discontinuation, as well as the use of corticosteroids and supportive care. Additional investigations to rule out other contributing causes, such as infection, pulmonary embolism, and congestive heart failure, should be completed, and consultation with respiratory specialists can be considered in more complex cases. When clinical and radiographic features favor a diagnosis of pneumonitis, the suspected offending drug is typically held, and corticosteroids are promptly initiated [95]. Symptomatic pneumonitis is usually treated with oral prednisone dosed at 1 mg/kg/day [96]. For severe cases, intravenous methylprednisolone may be administered at 1–2 mg/kg/day [97,98]. Corticosteroids are generally effective for treating anti-cancer therapy-induced pneumonitis, with clinical improvements observed in over 80% of patients with immune checkpoint inhibitor-induced pneumonitis [95].

Management of steroid-refractory treatment-related pneumonitis is not well characterized for many anti-cancer therapies, including novel agents such as ADCs and BsAbs. The most well-defined treatment guidelines exist for the management of immune checkpoint inhibitor-associated adverse events, including immune-related pneumonitis [95,99,100]. Typically, if ICI-related pneumonitis does not improve after 48 hours of corticosteroid therapy, it is considered steroid-refractory [101]. In these cases, additional immunosuppressive therapies, such as infliximab, mycophenolate mofetil, intravenous immune globulin (IVIG), or cyclophosphamide, may be considered [102]. Small studies suggest that tocilizumab might be a potential option to manage immune-related toxicities, including pneumonitis, although further studies are needed [103,104,105]. Supportive measures, such as supplemental oxygen and mechanical ventilation, may also be used. Overall, despite these options, there is no standard treatment for steroid-refractory pneumonitis. Clinical trials are ongoing to investigate personalized approaches for patients experiencing pneumonitis (e.g., NCT05455034). For patients who recover from acute symptoms of pneumonitis, long-term management of permanent fibrotic changes within the lung might be challenging. In this setting, treatment strategies are limited and include optimization of other underlying lung diseases (e.g., COPD), as well as supportive care, oxygen therapy, and consideration of pulmonary rehabilitation in patients with good cancer control and no other contraindications.

There are various factors that impact decision making regarding treatment rechallenge following drug-induced pneumonitis, including patient-related factors, severity of toxicity, and recovery of symptoms, among others. Typically, patients should have resolution of symptoms (≤grade 1) prior to consideration of rechallenge. Most guidelines do not recommend restarting the offending agent in those who experience grade 3 (e.g., severe symptoms that limit self-care or require oxygen or hospitalization) or higher pneumonitis [95,99,100,106]. A meta-analysis evaluating ICI rechallenge in NSCLC reported that in patients with ICI discontinuation due to irAEkhjkhjkjk, the incidence of grade 3/4 irAEs was lower with rechallenge compared to initial ICI treatment (8.6% vs. 17.8%, *p* = 0.001) [107]. However, recurrence rates for colitis, hepatitis, and pneumonitis appeared to be higher after ICI rechallenge compared with other irAEs [108]. Furthermore, studies suggest that approximately one-third of patients might develop recurrent pneumonitis with ICI rechallenge [108]. Therefore, there is a need to better identify which patients might be safely rechallenged with an ICI following drug-induced pneumonitis. An ongoing clinical trial aims to evaluate the safety of ICI retreatment with or without the addition of prednisone in NSCLC patients who discontinued prior ICI therapy due to immune-related toxicity (NCT03847649). In terms of retreatment with targeted therapies, a meta-analysis of *EGFR*-mutant NSCLC patients who developed pneumonitis from EGFR TKIs showed that approximately 1.13% of patients retreated with an EGFR TKI developed recurrent pneumonitis, with perhaps a higher incidence (up to 3%) with the 3rd generation EGFR TKI osimertinib; however, further detailed statistical comparison between groups was not feasible due to a small number of cohorts in this study [15]. Interestingly, studies suggest that Japanese patients appear to have a higher incidence of pneumonitis with up to five times higher odds compared to other groups; therefore, patient factors should be considered when evaluating the risk of pneumonitis with EGFR TKI retreatment. There is limited information on the safety of rechallenge with ADCs or BsAb therapies. In fact, for most BsAbs or ADCs currently used in clinical practice for NSCLC, permanent discontinuation of the agent is generally recommended after the development of symptomatic pneumonitis. For instance, in the product monograph for amivantamab, permanent discontinuation of the drug is recommended for confirmed ILD/pneumonitis of any grade [109]. Similarly, in the product monograph for T-DXd, permanent discontinuation of the ADC is recommended for symptomatic (grade 2 or higher) ILD/pneumonitis, whereas rechallenge might be considered in patients with asymptomatic (grade 1) ILD/pneumonitis after resolution of symptoms [110]. Further long-term evaluation of these novel agents will be needed to better understand the safety of rechallenge with ADCs or BsAbs.

Given the potential morbidity and mortality associated with treatment-related pneumonitis, it is imperative to find novel agents that might help prevent or decrease the incidence or severity of this complication. Various experimental approaches have been postulated but remain under investigation. For example, pentoxifylline and vitamin E have shown some benefits in the treatment of superficial radiation-induced fibrosis [111,112,113], with single-center studies suggesting possible impact on radiation-related pneumonitis [114], and case reports indicating a potential impact on bleomycin-related lung toxicity [115]. While case reports and retrospective analyses have suggested that the anti-fibrotic agent pirfenidone might have benefit in the management of bleomycin-induced lung injury [116,117] or immune checkpoint inhibitor pneumonitis [118,119], all of the reported cases were treated with high dose steroids and withholding the offending therapy, making it impossible to conclude that the antifibrotic therapy led to the observed improvement. Meanwhile, clinical studies evaluating the effect of various agents to prevent treatment-induced pneumonitis are ongoing (e.g., NCT06860542, NCT06634056). A better understanding of the pathophysiology associated with the development of treatment-related pneumonitis will aid in the elucidation of preventative strategies, and further studies are required to evaluate novel agents that might be utilized to help prevent anti-cancer treatment-related pneumonitis.

## 5. Discussion

Lung cancer management has evolved significantly over the last couple of decades with the incorporation of a more targeted approach to treatment based on the presence of molecular alterations or proteins expressed on tumor cells or immune cells. While TKIs and ICIs have been incorporated into routine clinical practice, novel therapies such as ADCs and BsAbs have shown the potential to further improve outcomes for lung cancer patients. However, understanding the unique toxicities associated with these agents is imperative.

ADCs have been associated with drug-induced pneumonitis, with variability in the incidence of pneumonitis related to each agent. Early-phase clinical trials of T-DXd, a HER2-directed ADC, have suggested a dose-dependent effect on the incidence of treatment-related pneumonitis. In the phase 2 DESTINY-Lung01 trial evaluating T-DXd in HER2-overexpressing NSCLC patients, there was an incidence of 20% for all-grade pneumonitis, with grade 3 or higher events in 6% of patients, including three deaths attributable to drug-related pneumonitis in patients treated with the 6.4 mg/kg dose (cohort 1) [41]. In contrast, of the 41 HER2-overexpressing NSCLC patients treated with 5.4 mg/kg of T-DXd (cohort 1A), only 7.3% were found to have pneumonitis, with 4.9% grade 3 or higher events, including two deaths. Moreover, in the phase 2 DESTINY-Lung02 study comparing the 5.4 mg/kg vs. 6.4 mg/kg dosing of T-DXd in *HER2*-mutant NSCLC patients, there was a higher incidence of drug-related pneumonitis (12.9% vs. 28.0%, respectively) with the higher dose [22]. Similar dose-dependent effects have been observed with Dato-DXd (TROP2 ADC) in the phase 1 TROPION-PanTumor01 trial, with a reported incidence of adjudicated drug-related ILD of 10% (n = 5) in the 4 mg/kg group, 6% (n = 3) in the 6 mg/kg group, and 13.8% (n = 11) with 8 mg/kg [25]. Importantly, all three grade 5 events occurred in the higher 8 mg/kg dose cohort [25].

Apart from dose, other variables may potentially influence the rate of pneumonitis with ADCs, including the drug structure, which can play a key role in defining the toxicity profile of this class of drugs. The DESTINY-Breast03 phase 3 randomized study compared T-DXd (5.4 mg/kg) and T-DM1 (3.6 mg/kg) in 524 patients with HER2-positive breast cancer. The incidence of drug-related pneumonitis was 10.5% (n = 27 patients: 7 grade 1 events, 18 grade 2, and 2 grade 3) for the patients treated with T-DXd compared to 1.9% (n = 5 patients: 4 grade 1 events and 1 grade 2 event) for patients receiving T-DM1 [40]. Although they are directed against the same antigen, T-DXd and T-DM1 are characterized by important differences in the linker and payload. T-DXd combines trastuzumab with a highly potent topoisomerase I inhibitor payload (DXd) via a cleavable linker. This payload is designed to be released inside the cancer cells, but its membrane-permeable nature allows it to diffuse into surrounding tissues, leading to a “bystander effect” where adjacent non-target cells can also be affected [42,43,44]. The high drug-to-antibody ratio (8:1) of T-DXd ensures a substantial amount of the cytotoxic payload is delivered to the tumor site. However, this also increases the likelihood of off-target effects as the released DXd can affect nearby healthy tissues, including the lungs [22,43,44]. In contrast, T-DM1 has a lower DAR of 3.5:1 and uses a microtubule inhibitor payload that is less likely to diffuse into surrounding tissues, resulting in fewer off-target effects. Additionally, the linker used in T-DXd is designed to be cleaved by lysosomal enzymes, which can lead to the release of the cytotoxic drug in non-target tissues, including the lungs. This mechanism contrasts with T-DM1, which uses a non-cleavable linker, reducing the likelihood of off-target release of the cytotoxic agent [42,44]. Importantly, a lower rate of lung toxicity for patients treated with T-DM1 has been confirmed for NSCLC patients, with 0 cases of pneumonitis observed in 49 patients with HER2-positive NSCLC treated with T-DM1 in a phase 2 study [45].

Many tumor-associated antigens that are targeted by ADCs are also expressed on normal tissues, including within the lung. It is unclear what the impact of normal tissue expression of these antigens in the lung might have on the risk of pneumonitis in lung cancer patients. A study by Desai et al. evaluated expression of common ADC targets (e.g., TROP2, HER2, HER3, MET) in both tumor and normal lung tissue and found higher RNA expression of these targets in the tumor compared to normal lung tissue [120]. There was moderate intensity (2+) IHC staining for HER2, HER3, and CEACAM5 (but weak MET staining) on alveolar cells, as well as moderate to high intensity IHC staining for HER2, HER3, CEACAM5, and MET in tissue macrophages [120]. However, despite the presence of these targets within normal lung tissue, there was no clear correlation with target antigen expression and incidence of drug-induced ILD [120], thereby highlighting the need for further studies into the impact of normal tissue target expression on drug-related toxicities.

The primary tumor type might also affect the incidence of ADC-associated pneumonitis. T-DXd has been evaluated in multiple tumor types, including HER2-positive breast cancer, gastric cancer, and NSCLC. Interestingly, the incidence of drug-induced pneumonitis at the higher 6.4 mg/kg dose of T-DXd was reported in 10% (n = 12: three grade 1, six grade 2, two grade 3, one grade 4, and no grade 5 events) of HER2-positive gastric cancer patients in the DESTINY-Gastric01 trial [121], while 28% (n = 14: four grade 1, nine grade 2, one grade 5) of *HER2*-mutant metastatic NSCLC patients had adjudicated drug-related ILD in the phase 2 DESTINY-Lung02 trial [41]. It is possible that the higher incidence of pneumonitis in NSCLC patients might be due to increased risk of off-target bystander effects with the primary tumor location within the lung; however, further research is needed to determine the exact etiology for this higher risk compared to other tumor types.

There is ongoing development of new antibody-drug conjugates in lung cancer, including ADCs directed at immune cell targets such as PD-L1 and B7-H3. Early-phase studies have evaluated B7-H3 ADCs in extensive-stage small cell lung cancer patients [122,123], and PD-L1 ADCs are being evaluated in phase 1 studies of solid tumors [124]. It remains unknown at this time if the incidence of pneumonitis with these novel ADCs directed at immune cell targets will be comparable to the rates of pneumonitis seen with other immune checkpoint inhibitor therapies currently utilized in clinical practice.

BsAbs are molecules that consist of two distinct antigen-binding sites, which can engage both tumor-associated antigens and/or immune cell receptors. The BsAb amivantamab (targeting tumor-associated antigens EGFR and cMet) appears to have similar rates of pneumonitis compared to other EGFR inhibitors used in NSCLC treatment. In the Phase I CHRYSALIS trial evaluating amivantamab in EGFR exon 20 insertion-mutated NSCLC, there was an incidence of 3.3% for all-grade pneumonitis, with 0.7% of patients experiencing grade 3 or higher events [30,64]. In the MARIPOSA trial, amivantamab plus lazertinib was compared to osimertinib (the current standard of care third-generation EGFR TKI) in the first-line treatment of *EGFR*-mutant NSCLC, and the incidence of pneumonitis or ILD was similar, with an incidence of 3% in both groups, including grade 3 or higher events in 1% of patients in both treatment arms [31]. A retrospective analysis by Schoenfeld et al. suggested an increased risk of pneumonitis (e.g., 9.7% incidence of grade 3 pneumonitis) with the use of osimertinib sequentially following a PD-(L)1 immune checkpoint inhibitor; interestingly, this risk of pneumonitis was specific to osimertinib and not observed with other EGFR TKIs (e.g., the first-generation TKI erlotinib or second-generation TKI afatinib) [125]. Further evaluation is needed to determine if sequencing of novel therapies such as ADCs or BsAbs after standard treatments, such as immune checkpoint inhibitors, might influence the risks of pneumonitis in lung cancer patients.

Bispecific T-cell engagers are a type of BsAb, which bind both a tumor-associated antigen and CD3 on T-cells. Tarlatamab is a bispecific T-cell engager (directed towards DLL3 and CD3) that has been studied in extensive-stage SCLC and has shown promising activity in early-phase trials. Interestingly, there have been low rates of pneumonitis reported (approximately <5%) in phase 1/2 studies of tarlatamab, which appears to be less than the incidence of pneumonitis typically seen with PD-L1/PD-1 inhibitors in lung cancer patients. Furthermore, novel bispecific antibodies targeting two distinct immune checkpoints (e.g., anti-PD1/anti-TIGIT, anti-PD1/anti-CTLA4, and anti-PD1/anti-LAG3 bispecific antibodies) are in clinical development and being evaluated in clinical trials. The risks of pneumonitis related to these agents have not been fully reported and given known risks of pneumonitis with PD-(L)1 and CTLA4 inhibitors, it will be interesting to see if these agents have similar toxicity profiles compared to current standard treatments. Overall, a better understanding of the differences in the mechanism of action between these novel agents and the impact on the development of immune-mediated pneumonitis is needed and should be elucidated in future studies.

Predictive biomarkers are critical tools to identify upfront patients at higher risk of toxicity and enable personalized monitoring and prevention strategies. Several non-invasive biomarkers, including radiomic features and blood-based analyses such as TCR metrics, immune-cytokines, and autoantibody panels, are emerging as potential factors for risk stratification in patients treated with radiotherapy and immunotherapy. While no data are available yet to confirm their utility in predicting pneumonitis in patients treated with ADCs and BsAbs, the known immunoregulatory properties of these novel classes of agents suggest potential applicability of similar biomarkers in this setting. Results from clinical trials currently ongoing will be needed to confirm their validity.

Similarly, optimal strategies for the treatment of pneumonitis from ADCs and BsAbs, particularly steroid-refractory cases, still need to be defined. Dedicated studies are needed to confirm the utility of immunomodulatory approaches such as IVIG or tocilizumab in this context.

Limitations of our work include the inherent selection bias associated with narrative reviews due to their non-systematic nature. Additionally, there was heterogeneity in the study populations, drug dosing, and variability in the definition and grading of pneumonitis, which limit the synthesis of findings and the generalizability of the conclusions.

## 6. Conclusions

Novel anti-cancer agents, including ADCs and BsAbs, represent new therapeutic options for patients with lung cancer; however, as with other oncologic treatments, they are associated with a risk of pneumonitis. Given the potential for increased risk of severe and fatal episodes of pneumonitis in lung cancer patients, the identification of reliable predictive biomarkers and a better understanding of the mechanisms underlying this toxicity are urgently needed to ensure prompt identification and proper management of pneumonitis. Correlative analysis from the multiple clinical trials currently ongoing and results from studies investigating prevention strategies and new treatment approaches for steroid-refractory pneumonitis will be a valuable resource to reduce the rate of complications and fatal cases of pneumonitis and overall improve the care of lung cancer patients.

## Figures and Tables

**Figure 1 cancers-17-02522-f001:**
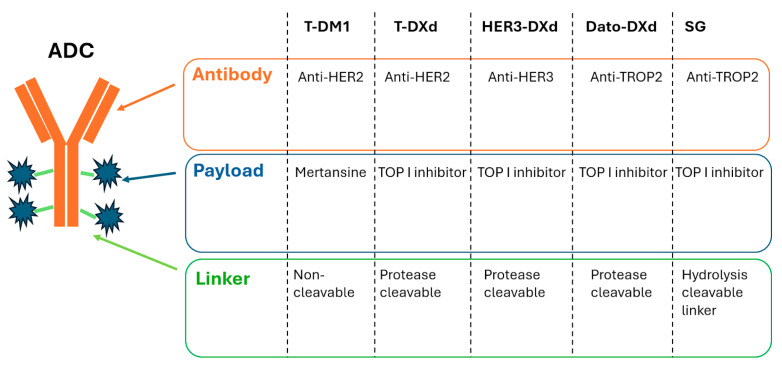
General antibody–drug conjugate (ADC) structure and specific components of main ADCs currently approved and in development for the treatment of lung cancer patients.

**Figure 2 cancers-17-02522-f002:**
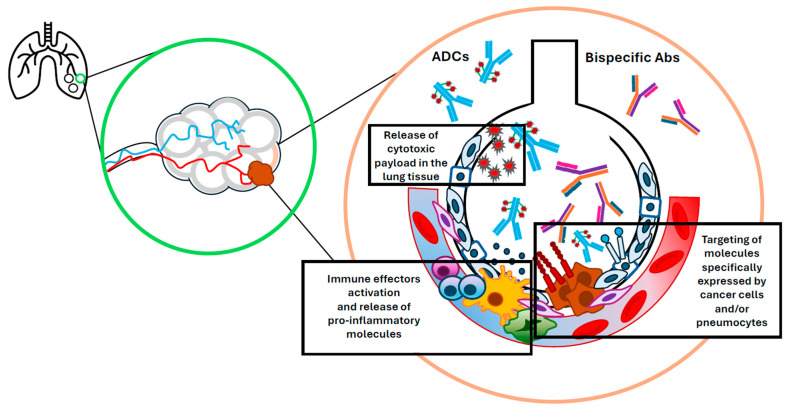
Mechanisms contributing to pneumonitis in cancer patients treated with antibody-drug conjugates and bispecific antibodies.

**Figure 3 cancers-17-02522-f003:**
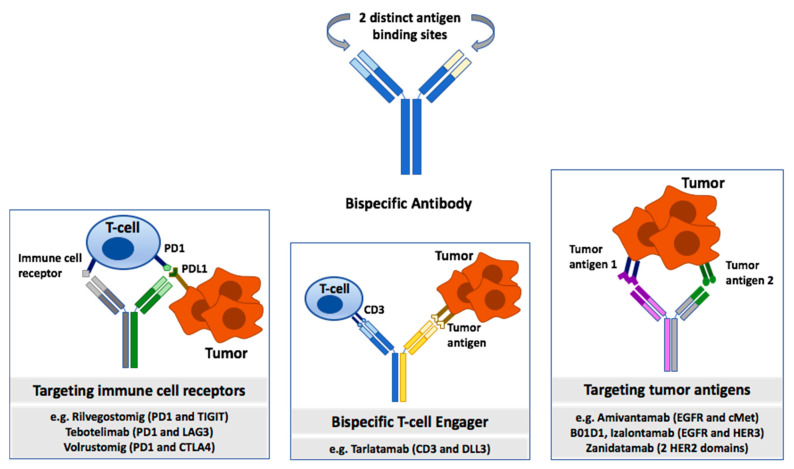
Bispecific antibodies with distinct antigen binding sites targeting tumor-associated antigens and/or immune cell targets.

**Table 1 cancers-17-02522-t001:** Treatment-induced pneumonitis in cancer patients receiving conventional therapies.

Treatment Type	Study Type	No. of Patients	Patient Population	Incidence of Pneumonitis		
				All Grade	Grade 3 or Higher	Grade 5
PD-1 and PD-L1 inhibitors	Systematic review and meta-analysis PMID 28499515 [12]	Subgroup 1: 3232 Subgroup 2: 1806	Subgroup 1: PD-1 inhibitors Subgroup 2: PD-L1 inhibitors	Subgroup 1: 3.6% (95% CI: 2.4–4.9%) Subgroup 2: 1.3% (95% CI: 0.8–1.9%)	Subgroup 1: 1.1% (95% CI: 0.6–1.7%) Subgroup 2: 0.4% (95% CI: 0–0.8%)	Not reported
EGFR-TKIs	Meta-analysis PMID 30089596 [15]	15,713	Subgroup 1: NSCLC patients without prior exposure to EGFR-TKIs Subgroup 2: NSCLC patients with prior exposure to EGFR-TKIs	Subgroup 1: 1.12% (95% CI: 0.79–1.58%) Subgroup 2: 1.13% (95% CI:0.40–3.15%)	Subgroup 1: 0.61% (95% CI: 0.40–0.93%) Subgroup 2: 0.49% (95% CI: 0.21–1.11%)	Subgroup 1: 0.20% (95% CI: 0.11–0.38%) Subgroup 2: 0.16% (95% CI: 0.04–0.65%)
ALK-TKIs	Systematic review and meta-analysis PMID 37017467 [19]	4752	NSCLC patients	2.92% (95% CI: 1.79–4.27%)	1.42% (95% CI: 0.84–2.12%)	0.09% (95% CI: 0.00–0.28%)
Chemo-radiation	Systemic review and meta-analysis PMID 35717343 [20]	1788	Unresectable stage 3 NSCLC	Not reported	RCTs: 3.62% (95% CI: 1.65–6.21) Observational studies: 5.98% (95% CI: 2.26–12.91)	RCTs: 0.37% (95% CI: 0–2.78) Observational studies: 1.73% (95% CI: 0.53–4.33)

**Table 2 cancers-17-02522-t002:** Treatment-induced pneumonitis in patients receiving novel anti-cancer agents.

Drug or Therapy	Study Name and Type	No. of Patients	Drug Dose	Incidence of Pneumonitis			Median Time to Onset of Pneumonitis	Median Duration of Pneumonitis
				All Grade	Grade 3 or Higher	Grade 5		
Trastuzumab deruxtecan (ADC)	DESTINY-Lung01 [21] (phase 2 study) PMID 38547891	Cohort 1: 49 Cohort 1A: 41	Cohort 1: 6.4 mg/kg dose once every 3 weeks Cohort 1A: 5.4 mg/kg dose once every 3 weeks	Cohort 1: 20% Cohort 1A: 5%	Cohort 1: 6% Cohort 1A: 2%	Cohort 1: 6% Cohort 1A: 2%	Not reported	Not reported
Trastuzumab deruxtecan (ADC)	DESTINY-Lung02 [22] (phase 2 study) PMID 37694347	Arm 1: 102 Arm 2: 50	Arm 1: 5.4 mg/kg dose once every 3 weeks Arm 2: 6.4 mg/kg dose once every 3 weeks	Arm 1: 12.9% (95% CI: 7.0–2.10) Arm 2: 28.0% (95% CI: 16.2–42.5%)	Arm 1: 2% Arm 2: 2%	Arm 1: 1% Arm 2: 2%	Arm 1: 88.0 (range: 20–421) days Arm 2: 83.5 (range 36–386) days	Not reported
Patritumab-deruxtecan (ADC)	HERTHENA-Lung01 [23] (phase 2 study) PMID 37689979	225	5.6 mg/kg once every 3 weeks	5.3%	1.3%	0.4%	53 (range: 9–230) days	Not reported
Patritumab-deruxtecan (ADC)	HERTHENA-Lung02 [24] (phase 3 study)	~293	5.6 mg/kg once every 3 weeks	5%	1%	>1%	Not reported	Not reported
Datopotamab deruxtecan (ADC)	TROPION-LUNG01 [25] (phase 3 study) PMID 39250535	299	6 mg/kg once every 3 weeks	8.8%	2.7%	None	52 days	Not reported
Datopotamab deruxtecan (ADC)	TROPION-PanTumour01 [26] (phase 1 study) PMID 37327461	6 mg/kg dose: 50 8 mg/kg dose: 80	0.27–10 mg/kg once every 3 weeks during escalation 4, 6, or 8 mg/kg once every 3 weeks during expansion	6 mg/kg dose: 6% 8 mg/kg dose: 13.8%	6 mg/kg dose: 2% 8 mg/kg dose: 5%	6 mg/kg dose: none 8 mg/kg dose: 3.75%	Not reported	Not reported
Sacituzumab govitecan (ADC)	TROPiCS-03 [27] (phase 2 study) PMID 39083724	~40	10 mg/kg once on day 1 and day 8 of a 21-day cycle until disease progress, unacceptable toxicity, study withdrawal, or death	None reported	None reported	None reported	n/a	n/a
Tarlatamab (BsAbs)	DeLLphi-300 [28] (phase 1 study) PMID 36689692	107	Dose exploration (0.003–100 mg every 2 weeks) Expansion dose: 100 mg	4.7%	1.8%	0.9%	Not reported	Not reported
Tarlatamab (BsAb)	DeLLphi-301 [29] (phase 2 study) PMID 39876075	220	10 mg every 2 weeks	1.4%	1.4%	None reported	Not reported	Not reported
Zanidatamab (BsAb)	Meric-Bernstam et al. [30] (phase 1 study) PMID 36400106	46	3 + 3 dose escalation (5–30 mg/kg every 1, 2, or 3 weeks)	None reported	None reported	None reported	n/a	n/a
Amivantamab (BsAb)	CHRYSALIS trial [31] (phase 1 study) PMID 34339292	352	1050 mg (1400 mg, ≥80 kg) once a week for the first 4 weeks and then once every 2 weeks starting at week 5	3.3%	0.7%	None reported	Not reported	Not reported
BL-B01D1 (BsAb)	Ma et al. [32] (phase 1 study) PMID 38823410	Total: 195 NSCLC: 113	2.5–3.5 mg/kg (days 1 and 8 every 3 weeks) or 4.5–6.0 mg/kg (once every 3 weeks)	0.5%	Not reported	Not reported	Not reported	Not reported
Izalontamab (BsAb)	Xue et al. [33] (phase 1 study) PMID 40260627	60	3 + 3 dose escalation (nine dose levels from 0.4 to 28.0 mg/kg) and dose expansion (five dose levels from 6.0 to 21.0 mg/kg). Izalontamab was administered weekly or every 2 weeks in a 4-week cycle	None reported	None reported	None reported	n/a	n/a

**Table 3 cancers-17-02522-t003:** Emerging biomarkers of treatment-induced pneumonitis in lung cancer patients.

Biomarker Category	Predictive Biomarker(s) Associated with Pneumonitis	Associated Agent(s)	References
Tissue-Based	Single-Nucleotide Polymorphism	Radiotherapy	[75,76,77,78,79]
	PD-L1 expression	ICIs	[74]
Blood-Based	Cytokines	Radiotherapy, ICIs	[80,81,82]
	Specific HLA subtypes	ICIs	[79,83,84]
	Autoantibodies	ICIs	[85,86]
	TCR metrics	ICIs	[87]
	White Blood Cell Counts	ICIs	[88]
Imaging-Based	Radiomics	ICIs, Radiotherapy	[89,90,91,92]

## Data Availability

No new data were generated or analyzed in this study. All data discussed are available in the published literature cited in this review.

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
