# Peer review of "Balancing Innovation and Safety: Prediction, Prevention, and Management of Pneumonitis in Lung Cancer Patients Receiving Novel Anti-Cancer Agents"

_cancers, 2025, doi:10.3390/cancers17152522_

Round 1
Reviewer 1 Report
Comments and Suggestions for Authors
In this study the authors address innovation and safety: prediction, prevention and management of pneumonitis in lung cancer patients receiving novel anticancer agents. Despite this some points should be addressed before publishing.
Abstract should be be adhere the standard abstract writing background, aim methods, results and conclusions.
Introduction should be contains information about epidemiology of lung cancer as global and regional health problems ,it's Risk factors prediction and detection. Also, the types of lung cancer in detail. Moreover, add information about advantages and disadvantages of novel and advanced anticancer agents compared to classical ones.As well, add details about prediction and detection of anticancer induced pneumonitis. Please, add the study rationale and aims at the end of the introduction section.
Methods, please add section about the type of the study (Narrative review, systematic, etc). Moreover, add information about software and academic research engine used in data collection such as Pubmed, Google scholar, Scopus Medline and others used in the manuscript preparation.
Please, add table compare between different novel anticancer used in treatment of lung cancer and management of thier pneumonitis as major health problem.
Please, add table indicating biomarkers for prediction and detection of novel anticancer induced pneumonitis.
Please, add scheme illustrated the management of novel anticancer induced pneumonitis
Please, support your discussion with similar published documents. Moreover, discuss the study limitations.
Please check the manuscript for misuse of acronyms as well as long sentences or paragraphs without references citation.
Please check the references list for 2025 citation dated.
Comments on the Quality of English Language
Please check the manuscript for minor grammar errors and syntax.
Author Response
Reviewer 1: In this study the authors address innovation and safety: prediction, prevention and management of pneumonitis in lung cancer patients receiving novel anticancer agents. Despite this some points should be addressed before publishing.
- Abstract should be be adhere the standard abstract writing background, aim methods, results and conclusions.
- Response: Thank you for the comment. The suggested abstract subheadings (background, methods, results, conclusion) are typically recommended for an ‘original research article’ or ‘systematic review’. Given the current article is a narrative review, we did not feel that these subheadings were necessary to reflect the content of the article. Therefore, we have not made any changes to the format of the abstract.
- Introduction should be contains information about epidemiology of lung cancer as global and regional health problems,it's Risk factors prediction and detection. Also, the types of lung cancer in detail.
- Response: As suggested by the reviewer, we have added the following text to the introduction (see page 2, lines 53-65):
“Lung cancer is the leading cause of cancer-related deaths worldwide, with around 2.48 million new cases and 1.8 million deaths annually. In Canada, it is estimated that 32,100 individuals will be diagnosed with lung cancer in 2024, and approximately 20,700 Canadians will die from the disease. Lung cancer is primarily classified into non-small cell lung cancer (NSCLC) and small cell lung cancer (SCLC). NSCLC (e.g. adenocarcinoma, squamous cell carcinoma, and large cell carcinoma) accounts for around 85% of lung cancer cases. On the other hand, SCLC is a more aggressive form of lung cancer, which tends to spread rapidly to distant organs, and accounts for approximately 15% of lung cancer cases. Major risk factors for both NSCLC and SCLC include cigarette smoking, second-hand smoke, occupational exposure, air pollution, and genetic susceptibility. Currently, lung cancer is diagnosed through imaging and pathological assessment of tissue biopsies. Low-dose CT scans are typically used for early detection in high risk individuals, while biopsy remains the gold standard for a definitive diagnosis.”
Moreover, add information about advantages and disadvantages of novel and advanced anticancer agents compared to classical ones.
- Response: Thank you for the comment. We have highlighted advantages and disadvantages (as they pertain to pneumonitis risk) for each of the novel anti-cancer agents within the body of the text. Therefore, we have not added extra text to the introduction.
As well, add details about prediction and detection of anticancer induced pneumonitis.
- Response: Thank you for the comment. We have a section on predictive biomarkers of treatment-related pneumonitis that addresses this reviewer comment. This information has also been summarized in Table 3 (as addressed in reviewer 1, comment #5 below).
Please, add the study rationale and aims at the end of the introduction section.
- Response: We have added the following text to the introduction, as recommended by the reviewer (see page 3, lines 95-99):
“This narrative review aims to 1) examine the incidence of pneumonitis associated with conventional and novel anti-cancer therapies, particularly ADCs and BsAbs, 2) elucidate potential pathophysiological mechanisms underlying pneumonitis risk, and 3) evaluate emerging predictive biomarkers, which could help minimize future pneumonitis risk for lung cancer patients.”
- Methods, please add section about the type of the study (Narrative review, systematic, etc). Moreover, add information about software and academic research engine used in data collection such as Pubmed, Google scholar, Scopus Medline and others used in the manuscript preparation.
- Response: Thank you for the comment. As this article is a narrative review (i.e. not a systematic review), a detailed methods section was not included. However, we have added the following text to the introduction to address the reviewer’s comments regarding the literature search conducted (see page 3, lines 99-101)
“A health sciences librarian was consulted, and a literature search was completed utilizing the Medline, Embase, PubMed and Google Scholar databases.”
- Please, add table compare between different novel anticancer used in treatment of lung cancer and management of their pneumonitis as major health problem.
- Response: Thank you so much for your suggestion. We have highlighted the differing rates of pneumonitis associated with traditional and novel anticancer therapies in Tables 1 and 2. As discussed in the management section, there are currently no distinct treatment guidelines based on the specific causative agent for novel therapies such as bispecific antibodies and antibody-drug conjugates. While more structured recommendations exist for immune checkpoint inhibitor–related pneumonitis, the management of toxicities from novel agents (including bispecific antibodies and antibody-drug conjugates) is often extrapolated from those guidelines. One of the main goals of this work is to emphasize how much remains unknown and to underscore the need for further research to support the development of agent-specific treatment recommendations.
- Please, add table indicating biomarkers for prediction and detection of novel anticancer induced pneumonitis.
- Response: Thank you for this suggestion. We have added Table #3 to summarize the biomarkers discussed in the manuscript
- Please, add scheme illustrated the management of novel anticancer induced pneumonitis
- Response: Thank you for this comment. As stated above (in response to reviewer 1, comment #4), there is no clear evidence yet to guide a specific approach to management for patients experiencing pneumonitis from novel anticancer agents such as bispecific antibodies and antibody-drug conjugates.
- Please, support your discussion with similar published documents. Moreover, discuss the study
- Response: Thank you for the comment. A detailed limitations section is typically not indicated in a narrative review; however, we have added a brief statement at the end of the discussion section to address this reviewer comment (see page 12, lines 595-598).
“Limitations of our work include the inherent selection bias associated with narrative reviews due to their non-systematic nature. Additionally, there was heterogeneity in the study populations, drug dosing, and variability in the definition and grading of pneumonitis, which limit synthesis of findings and generalizability of the conclusions.”
- Please check the manuscript for misuse of acronyms as well as long sentences or paragraphs without references citation.
- Response: Thank you, we have checked acronyms throughout the manuscript. We have updated the list of abbreviations section to include all of the acronyms used in the manuscript. We have also made further revisions to the text throughout the manuscript to address some grammatical concerns.
- Please check the references list for 2025 citation dated.
- Response: It is unclear what the reviewer question is. We would appreciate some clarification for this comment. Thank you.
Reviewer 2 Report
Comments and Suggestions for Authors
Pneumonitis is an adverse event with a rising incidence among patients receiving systemic treatments for lung cancer, significantly affecting both therapy outcomes and quality of life. This underscores the relevance of this review and the unmet needs it highlights.
The article is well written and structurally sound. The main limitations are found in the tables; they would benefit from restructuring, such as condensing some columns (for example, combining drug or therapy name with drug type, or merging study type, study name, and PMID) and removing others, like patient population.
The summary and abstract effectively highlight the main topics of the article. The introduction would be strengthened by adding a final paragraph that reinforces the objectives of the review.
Regarding novel anticancer agents, consider updating data on tarlatamab (line 244-245). The phase 3 trial has been published.
On the topic of management consider adding a workout proposal for differential diagnosis.
Author Response
Reviewer 2: Pneumonitis is an adverse event with a rising incidence among patients receiving systemic treatments for lung cancer, significantly affecting both therapy outcomes and quality of life. This underscores the relevance of this review and the unmet needs it highlights.
- The article is well written and structurally sound. The main limitations are found in the tables; they would benefit from restructuring, such as condensing some columns (for example, combining drug or therapy name with drug type, or merging study type, study name, and PMID) and removing others, like patient population.
- Response: Thank you for the comment. As suggested by the reviewer, we have re-formatted the
- Table 1 – Removed study authors and drug type, as well as PMID column.
- Table 2 – Merged drug name and drug type; merged study name and study type; deleted study population and PMID columns
- The summary and abstract effectively highlight the main topics of the article. The introduction would be strengthened by adding a final paragraph that reinforces the objectives of the review.
- Response: Thank you for the comment. As addressed above in response to reviewer 1 comments, we have added the following text to the introduction (see page 3, lines 95-99):
“This narrative review aims to 1) examine the incidence of pneumonitis associated with conventional and novel anti-cancer therapies, particularly ADCs and BsAbs, 2) elucidate potential pathophysiological mechanisms underlying pneumonitis risk, and 3) evaluate emerging predictive biomarkers, which could help minimize future pneumonitis risk for lung cancer patients.”
- Regarding novel anticancer agents, consider updating data on tarlatamab (line 244-245). The phase 3 trial has been published.
- Response: We have included the recently published results from the phase 3 DeLLphi-304 trial, as recommended by the reviewer (see page 6, lines 266-270)
“In the recently published Phase 3 DeLLphi-304 trial, tarlatamab (10 mg every 2 weeks) was associated with an overall survival benefit compared to chemotherapy (13.6 months vs. 8.3 months) in the second-line treatment of extensive stage SCLC patients. There were no reported cases of pneumonitis observed in the 254 patients treated with tarlatamab; the most common adverse event was low grade cytokine release syndrome.”
- On the topic of management consider adding a workout proposal for differential diagnosis.
- Response: Thank you for the comment. The studies analyzed in this review were too heterogeneous in terms of treatment regimens and patient populations to support conclusions that could inform a definitive algorithm for differential diagnosis based on treatment type. The primary aim of our work was to highlight that meaningful differences may exist in the presentation and mechanisms of pneumonitis across novel therapeutic agents—differences that are not yet adequately addressed due to significant gaps in our current understanding. We believe that the development of such an algorithm will require prospective studies specifically designed to investigate the mechanisms and clinical features of pneumonitis associated with emerging cancer therapies.